# Probabilistic Neural Programmed Networks for Scene Generation

**Zhiwei Deng, Jiacheng Chen, Yifang Fu, Greg Mori**
Simon Fraser University
{zhiweid, jca348, yifangf}@sfu.ca, mori@cs.sfu.ca

## Abstract

In this paper we address the text to scene image generation problem. Generative models that capture the variability in complicated scenes containing rich semantics is a grand goal of image generation. Complicated scene images contain varied visual elements, compositional visual concepts, and complicated relations between objects. Generative models, as an analysis-by-synthesis process, should encompass the following three core components: 1) the generation process that composes the scene; 2) what are the primitive visual elements and how are they composed; 3) the rendering of abstract concepts into their pixel-level realizations. We propose PNP-Net, a variational auto-encoder framework that addresses these three challenges: it flexibly composes images with a dynamic network structure, learns a set of distribution transformers that can compose distributions based on semantics, and decodes samples from these distributions into realistic images.

## 1 Introduction

Powerful latent data representations should encode abstract, semantic concepts. Accompanying generative models to decode these rich representations into their varied realizations as data instances, along with learning of the latent representation directly from data represent a coveted guerdon of AI research. As such, the search for expressive, learnable latent encodings along with corresponding generation techniques has been a preoccupation of significant research effort.

Examples of data modeling tasks abound. Specifically in this work we consider that of image generation. We are particularly interested in the representation of high-level semantic concepts and hence focus on the depiction of a complex scene $x$, as illustrated in Fig. 1. Such scenes are composed of a variable number of objects, with individual properties and relations among them. Effective latent representations for these images need flexibility and compositionality.

Impressive strides in generative models for images have been made via a number of advances in continuous stochastic latent variable models under the variational auto-encoder (VAE) formalism [1, 2, 3]. We work within this formalism, in which a (conditional) prior $p(z|y)$ controls generation of an output $x$ under condition $y$ via a non-linear mapping $p_\theta(x|z)$. Recent work (e.g. [3]) has emphasized that the utility of a VAE hinges on its ability to capture useful information in the latent representation $z$, complementary to that in powerful decoder networks $p_\theta(x|z)$.

On the other hand, renewed interest in programmatic representations in AI have sought modular, generalized, compositional representations for high-level concepts [4, 5, 6, 7, 8]. This line of work uses dynamically programmable networks and has demonstrated impressive results at question answering, graphics-based image rendering, and synthesizing programs for computation or image generation. We build on these approaches to conduct learning of compositional, modular latent representations for scenes.

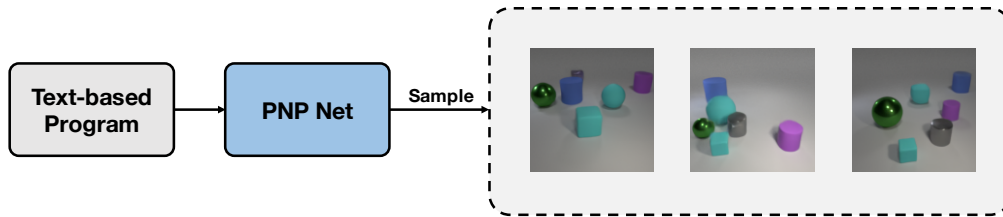

Figure 1: Generating a complicated scene with rich primitive concepts, compositionality and interactions is challenging. We use text based programs (can be derived from other forms of language, e.g. sentences) and propose PNP-Net, which is a powerful and flexible model for combining and generating scene images. Samples generated by our model are shown here.

The krux of the problem in learning latent data representations is this dichotomy between powerful decoder networks and the latent encoding. However, structured, complex scenes present an enticing opportunity for learning latent representations. While recent advances in generative modeling have produced exciting progress toward image generation, the successes are largely focused on images of a single object or hand-drawn symbol.

We leverage these lower-level image generation successes to build higher-level latent representations of semantic concepts. Modeling complex phenomena arguably requires a structured form to the prior $p(\boldsymbol{z}|\boldsymbol{y})$. We advocate for the use of a modular, compositional prior that permits learning disentangled representations that flexibly scale to variable numbers of, and relationships between, concepts in a scene. Specifically, we formulate a generic recursive form to the prior: $p(\boldsymbol{z}|\boldsymbol{y}) = \tau(p_1(\boldsymbol{z_1}|\boldsymbol{y}), \ldots, p_K(\boldsymbol{z_k}|\boldsymbol{y}))$, where each component of this prior is itself defined in a similar fashion. Consider the case of composing a scene of a shiny, red cylinder next to a matte, blue sphere. Priors for the properties of the objects such as $p_{shiny}(\boldsymbol{z}_{shiny}|\boldsymbol{y})$ and $p_{red}(\boldsymbol{z}_{red}|\boldsymbol{y})$ are composed via aggregation operators $\tau(\cdot, \ldots, \cdot)$ in a programmatic fashion, i.e. yielding a prior $p_{red,shiny}(\boldsymbol{z}_{red,shiny}|\boldsymbol{y}) = \tau(p_{red}(\boldsymbol{z_{red}}|\boldsymbol{y}), p_{shiny}(\boldsymbol{z_{shiny}}|\boldsymbol{y}))$ for the composite concept of red and shiny.

This paper describes Probabilistic Neural Programmed Networks (PNP-Net), a probabilistic modeling framework. PNP-Net combines the advantages of modular programmable frameworks with probabilistic modeling. The contributions of this paper include: 1) a set of visual elements and neural modules/programs for modifying the appearance of these visual elements; 2) integrating these probabilistic neural modules into the canonical VAE framework, generalizing the VAE by empowering it with reusable, composable, interpretable modules; and 3) demonstrating generalization ability for complicated scene understanding, including zero-shot learning of novel compositions.

This approach leads to compositional models of appearance that can be utilized across variable scenes. Learning is sample-efficient, in that shared properties (such as shiny cylinders and shiny spheres) can benefit from commonality among training samples. The model we present is effective at harnessing the strengths of powerful generative decoders, while encoding semantic properties. We demonstrate that we can learn compositional semantic priors that can capture the variability in complex scenes. These models outperform competing approaches on Color-MNIST and CLEVR-G image generation tasks, while yielding semantically-meaningful latent representations.

## 2 Related Work

This paper proposes a generative modeling approach by linking language elements to both semantic latent priors and functional symbolic networks. We draw links from the following related fields and review previous works in the following domains: generative modeling, compositional semantics, language in vision and disentangled representations.

**Generative modeling:** Image synthesis via generative models has received a flurry of renewed interest. Training in a generative adversarial framework [9] has been used for a variety of tasks. Promising results have been achieved for image-conditional tasks [10, 11]. These can be augmented with hierarchical models [12, 13, 14, 15] for maintaining image structure in the context of body pose.

Gregor et al. [16] advocate for a recurrent approach for image generation, with successive rounds of refinement in a generative process that remains close to the pixels. A push toward hierarchical

variants could include layers or successively more abstract information regarding image content. PixelCNN [17, 18] demonstrates the power of low-level architectures to model fine-scale pixel detail. Impressive (conditional) image generation results are achieved with a slate of variants that include recurrent architectures, convolutional approaches, and gated models.

**Generative models with learnable priors:** PixelVAE [2] utilizes hierarchical latent variables with auto-regressive structure, in line with a PixelCNN output pixel value decoding. Chen et al. [3] analyze the dichotomy between latent codes and powerful decoders. Approaches to narrow the decoder's view or auto-regressive latent codes can be used to encourage the VAE's latent code to store useful information. Hoffman [19] develops a Markov chain Monte Carlo algorithm for refining an initial variational approximation to the data likelihood.

**Programmatic representations and reasoning:** Neural Module Networks [5] dynamically construct neural network architectures by composition of modules. Wu et al. [8, 20] describe methods for extracting physical world representations for scenes and videos via structured building block components. Johnson et al. [21] develop symbolic modules for visual reasoning. Subsequent work [22] generates an image from a scene graph using graph convolution to decode object layouts.

SPIRAL [6] produces programs capable of generating images in an adversarial context. Reed and de Freitas [4] describe the Neural Programmer-Interpreter architecture. Building blocks including a key-value data store and composable modules that represent function calls and arguments are combined in a curriculum learning framework. Parisotto et al. [23] synthesize complex programs for input-output synthesis in a domain-specific language.

**Language and vision:** Words and pictures research has deep roots [24]. Recent work on image captioning typically uses encoder-decoder architectures (e.g. [25, 26]), i.e. learning a vector to represent the meaning of a word/sentence/description. There is also work combining word embedding with images. Karpathy et al. [27] understands language fragments with visual appearance. DeViSE [28] proposes a joint embedding between single word labels and images. But these works only use a single vector to represent an image and are not applicable to generative modeling. The realization of a concept covers infinitely many possible image instances, namely a distribution of images.

**Learning disentangled representations**: There has been significant work on learning disentangled representations. Kulkarni et al. [29] encourage VAE latent variables to focus on disentangled factors via a novel model structure and mini-batches with certain transformations active or inactive. InfoGAN [30] learns disentangled angles, lighting, etc. Reed et al. [31] applies visual differences to a new image and to make content changes. FVAE [32] learns disentangled representations for audience identities and times. In this work we aim for a more general image understanding with linking of visual appearance to abstract descriptions to more deeply understand semantics.

# 3   Model

In this section we describe how we can take the semantic description $y$ of a scene, and generate an image $x$ containing these concepts. Our proposed model has two core components: 1) a set of mapping functions $\tau(\cdot, \ldots, \cdot)$, which take either semantic concepts or a series of distributions as input, and generate distributions over the latent space capturing their combined meaning; 2) a probabilistic modeling framework which performs inference and learning using this latent space.

We work with a description $y$ that forms a tree structure. Construction of the latent representation $z$ proceeds with a bottom-up pass over the tree structure. Each node $i \in y$ has a type $t(i)$ and concept word $w(i)$. To each node we apply the appropriate mapping function $\tau_{t(i)}$ over it and its children, modulated by the content of the concept word $w(i)$. For example, a "describe" node with concept word "cube" would specify how to combine the child visual property "brown" with the object "cube" into a combined representation for a brown cube. Figure 3 shows an example tree structure.

This modular approach with mapping functions allows us to reuse aspects of our model and combine them to represent complex scenes. The framework permits varied mapping functions with different input arity and output dimension. In the following sections we elaborate on this model and provide the specific examples of the mapping functions we use (illustrated in Fig. 2).

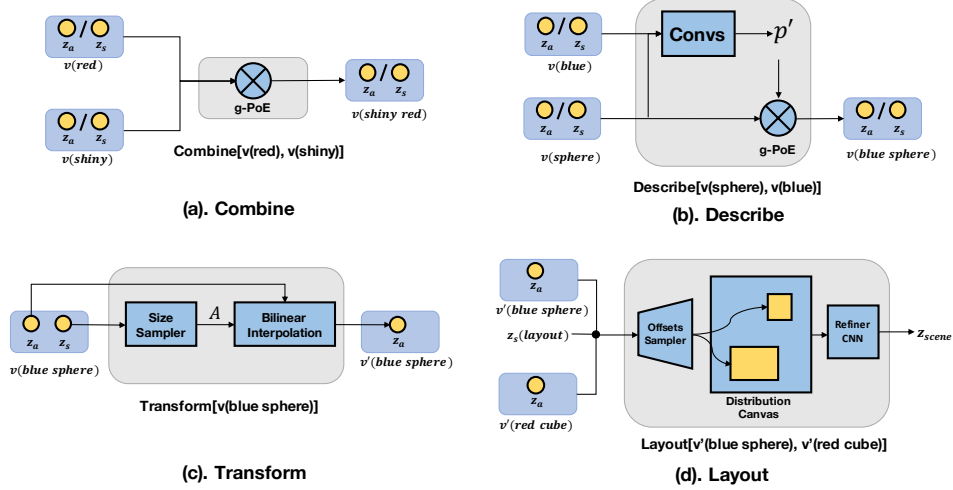

Figure 2: Core modules. a) **Combine** takes two distributions and generates a compound distribution as output. b) **Describe** models the "decorating" interaction between distributions and generated output distributions. c) **Transform** instantiates the size of a spatial appearance distribution, and performs bilinear interpolation on it to generate a new appearance distribution. d) **Layout** places two appearance distributions according to sampled offsets on a canvas and renders it by conv layers.

## 3.1 Probabilistic operators

Given abstract semantic concepts, there are many possible groundings in the visual domain. Assume there is a latent space which describes both the semantic concepts and visual data. Our goal is to define a set of reusable operators, which can be used to gradually compose a latent distribution $p(z|y)$ that fully describes the possible complicated scenes. We follow the standard variational autoencoder setup and use Gaussian distributions $p(z; \mu, \sigma)$ in the latent space, with $\mu$ and $\sigma$ as mean and variance respectively. To handle the background which is not described by the semantics, we keep a globally learned background mean $\mu$ and variance $\sigma$.

### 3.1.1 Concept mapping operator

Primitive concepts are the most basic elements, e.g. attributes (shiny, blue, metal, ...), objects (car, sphere, table, ...) and relations between objects (on top of, holding, ...). We first define a **concept mapping operator** $\tau_{concept}(w)$ which takes the concept word $w$, and generates distributions in latent space which can describe the properties of the concept. Ideally, the distribution should describe what the visual appearance variation is, and what the location/scale information is for that concept. We use the following coupled distributions to model the appearance and location/scale latent distribution:

$$\tau_{concept}(w) = \left[ p(z_a; f_\mu^a(w), f_\sigma^a(w)), p(z_s; f_\mu^s(w), f_\sigma^s(w)) \right] \quad (1)$$

here $f_\mu^{\cdot}$ and $f_\sigma^{\cdot}$ embed the word $w$ to its corresponding mean $\mu_{\cdot}$ and variance $\sigma_{\cdot}$ respectively. $z_a$ is a latent variable which characterizes the visual appearance for distribution of concept $w$, $z_s$ is the variable for the location/scale of $w$. Note that we design $\mu_a$ and $\sigma_a$ to be $C \times H \times W$ tensors with height $H$ and width $W$ for maintaining spatial appearance information, while $\mu_s$ and $\sigma_s$ are simply $C$-dim vectors encoding scale/location information.

### 3.1.2 Aggregation operators

Given the primitive concept latent distributions derived from the concept mapping operator, we now have the basic elements for composing objects and more complex scenes. In the rest of this section, we describe a set of aggregation mapping functions which operate on distributions. These operators take in intermediate distributions, and then generate output distributions based on the semantic meaning of the operator.

**Combine:** The combine operator combines attributes. It takes multiple attribute distributions, and aims to generate *compound latent distributions* which represent composite concepts "A and B".

For example, $\tau_{combine}(p_{shiny}, p_{metal})$ will output a distribution for *shiny metal*. This operator is defined separately for appearance and location/scale to process $p(z_a)$ and $p(z_s)$ in parallel due to the different properties of appearance and scale information. We use *Product of Experts (PoE)*[33, 34] for combining distributions and further propose a gated version of PoE (*g-PoE*) [1], which takes the parameters of distributions $p_i(\boldsymbol{z_i})$ and $p_j(\boldsymbol{z_j})$, and generates a set of gates $\{\boldsymbol{g}\}$ to control the information during combining. The operations of combine operator can be summarized by:

$$\tau_{combine}(p_i, p_j) = \frac{1}{Z} p(\boldsymbol{z}_i; \boldsymbol{g}_i^\mu \odot \boldsymbol{\mu}_i, \boldsymbol{g}_i^\sigma \odot \boldsymbol{\sigma}_i) p(\boldsymbol{z}_j; \boldsymbol{g}_i^\mu \odot \boldsymbol{\mu}_j, \boldsymbol{g}_i^\sigma \odot \boldsymbol{\sigma}_j) \tag{2}$$

where $\odot$ is element-wise product. Note that this operator can be reused repeatedly to combine more than two attributes.

**Describe:** The describe operator grounds attributes to an exact object. It models the interactions between attribute distributions and an object distribution in a content aware manner, considering that the way attributes affect an object depends on the properties of the target object. When composing the distribution for *red sphere*, it first takes the distributions of *red* and *sphere*, then generates a content aware "decorating" distribution $p'$, which is finally imposed on the distribution of sphere to get the distribution $p(\boldsymbol{z}|red, sphere)$. Note that this operator is also separately defined for appearance and location/scale to process $p(z_a)$ and $p(z_s)$ in parallel. We represent the describe module by:

$$\tau_{describe}(p_i, p_j) = p'(\boldsymbol{z}'; f_\mu(\boldsymbol{\mu}_i, \boldsymbol{\mu}_j), f_\sigma(\boldsymbol{\sigma}_i, \boldsymbol{\sigma}_j)) \otimes p_j(\boldsymbol{z}_j; \boldsymbol{\mu}_j, \boldsymbol{\sigma}_j) \tag{3}$$

where $\otimes$ represents *g-PoE*, and $f.(\cdot)$ is the function for combining mean or variance. Note that $f.(\cdot)$ is a CNN when the inputs are appearance distributions, while a MLP for scale/location distributions.

**Transform:** Scale invariance is critical to modeling visual content. The transform operator instantiates the size of an object by first sampling bounding box sizes from a scale distribution, and then performing bilinear interpolation on both mean and standard deviation to adapt the appearance distribution to varied sizes. More precisely, we first sample a scale tuple $s = (h, w)$ from $p(s|z_s), z_s \sim p(z_s)$, indicating the size of bounding box in latent space. Then the transformation matrix $\boldsymbol{A}$ can be defined as a diagonal matrix with scaling factor specified by $h$ and $w$. After the transformation, the re-sampled mean and variance will have new spatial size $h$ and $w$. The transform module is summaried by:

$$\tau_{transform}(p) = p(\boldsymbol{z}; bilinear(\boldsymbol{\mu}, \boldsymbol{A}), bilinear(\boldsymbol{\sigma}, \boldsymbol{A})) \tag{4}$$

**Layout:** The layout operator models the interactions between objects. Based on the semantic concepts such as *right-next-to*, *holding*, etc., it generates the positions for arranging latent distributions of two children nodes. Specifically, we first sample a tuple $l = (x, y)$ from $p(l|z_s)$, indicating the offsets between two children's bounding boxes in latent space, where $z_s \sim p(z_s)$ and $p(z_s)$ is generated by our *concept mapping operator* $\tau_{concept}$ taking the relation word of layout node. Then we place two masks on the background canvas guided by the offsets $l$ and the bounding box scales $s$ of children nodes. We then fill the distributions $p_i$ and $p_j$ from children nodes in the current canvas according to the masks. The rest of the canvas is filled up with background biases. The canvas size is set to the minimum size to cover all objects, or to the actual latent map size of whole image when layout is the root node. Denote the mean and variance of the final canvas as $\boldsymbol{\mu}_o^{ij}$ and $\boldsymbol{\sigma}_o^{ij}$, and let $f_\mu(\cdot)$ and $f_\sigma(\cdot)$ represent CNNs for further refining the final canvas, the final step of layout module can be expressed as:

$$\tau_{layout}(p_i, p_j) = p(\boldsymbol{z}; f_\mu(\boldsymbol{\mu}_o^{ij}), f_\sigma(\boldsymbol{\sigma}_o^{ij})) \tag{5}$$

## 3.2 Model formulation

Given the semantic description for image $\boldsymbol{x}$, semantic information $\boldsymbol{y}$ and operators $\{\tau(\cdot, ..., \cdot)\}$, we design a model that utilizes the set of operators to generate $x$ from $y$. We first map the semantic information into programmatic operations $\mathcal{P}$ which represent the generation process. Similar to [5], we use tree structures for $\boldsymbol{y}$ to represent the generation process, but note that our operators and method could be generalized to directed acyclic graph structures.

Following the generation process, a model should be able to gradually compose from primitive latent distributions to a full latent distribution $p(\boldsymbol{z}|\boldsymbol{y})$ describing a complex scene, where it samples a latent

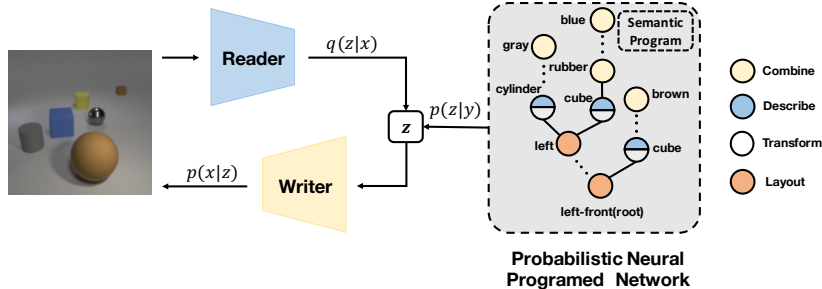

Figure 3: Our model takes a program derived from text concepts, performs compositional generation in the latent space $z$, then decodes the latent code to image space. During training, an encoder (Reader) is used as a proposal distribution for learning the latent space. **Describe** and **Transform** are put together in the figure to emphasize the fact that the application of a **Transform** module always follows another **Describe** module in our formulation.

$z$ as a possible valid scene, then renders it into an image (normally with a learnable general decoder). However, finding a latent space $\mathcal{Z}$ describing the observations (image $x$ and semantic concepts $y$) is known to be a hard problem, since calculating the true posterior distribution $p(z|x, y)$ is often intractable. We follow a standard variational inference framework, use a proposal distribution $q(z|x)$ to approximate such a latent space by minimizing the KL Divergence term $KL(q(z|x)||p(z|x, y))$, where $p(z|x, y)$ is the true posterior distribution. By re-writing the KL term, we get the following equation:

$$\log p(x|y) - KL(q(z|x)||p(z|x, y)) = \mathbb{E}_{z \sim q(z|x)}[\log p(z, x|y) - \log q(z|x)] \qquad (6)$$

$$= \mathbb{E}_{z \sim q(z|x)}[\log p(x|z)] - KL(q(z|x)||p(z|y)) \qquad (7)$$

$$= \mathbb{E}_{z \sim q(z|x)}[\log p(x|z)] - KL(q(z|x)||\mathcal{P}(\tau, y)) \qquad (8)$$

In the above equation, we formulate the prior distribution $p(z|y)$ as a learnable prior, and assume that $z$ should contain the full information for the decoder to reconstruct $x$. The learnable distribution is defined by the generation process $\mathcal{P}(\tau, y) = \tau(p_1(z_1|y), \dots, p_K(z_k|y))$. Since KL divergence is non-negative, we have $\log p(x|y) \geq \mathbb{E}_{z \sim q(z|x)}[\log p(x|z)] - KL(q(z|x)||\mathcal{P}(\tau, y))$. The right hand side is known as the *Evidence Lower Bound (ELBO)*, where the first term is the likelihood for reconstructing observations, and the second term is minimizing the KL divergence between a particular data sample and a complex semantic driven prior that could correspond to many possibilities. We optimize the *ELBO* to maximize the log likelihood for the conditional distribution $p(x|y)$. Along with the training process, all operators affiliated to the tree are updated through the second KL term. The flexibility of the operators in the prior permits a variety of groundings in the visual domain. The M-projection KL term for prior $p(z|x)$ encourages the scene prior to cover those groundings in the latent space. Note that to better train the **Transform** and **Layout** operators, we also use ground truth bounding boxes as auxiliary loss for learning size and offset samplers.

Through defining a small set of operators/modules, the prior distribution $p(z|y)$ conditioned on the semantic concepts can acquire much more flexibility. During generation, our model takes the programmatic process $\mathcal{P}$ and dynamically constructs a network $p(\cdot; \theta)$ by assembling and reusing the operators $\tau$. The network will finally generate the prior distribution $p(z|y; \theta)$, which is then mapped to possibilities of visual data $p(x|z)$ by a decoder function.

## 4 Experiments

We evaluate PNP-Net on the task of text to scene generation across a series of datasets and experimental settings with various complexity.

**Performance Measure**: Measuring the quality of samples for a generative model is challenging. Classifier-based scores have proven to be more directly relevant to the visual quality of generated samples. Inception score [35] measures the generated samples by using the final output of an Inception network trained for classification. However, most classifier based scores are designed for measuring

the quality of a whole image or single object. In contrast, we require a metric for measuring the quality of generated objects in a complex scene.

We propose to use the *mean average precision(mAP)* of a pre-trained detector to measure whether the conditional generated image samples possess desired semantic information. We modify the standard PASCAL VOC detection metric as follows: we measure the semantic correctness of conditional generation per class by considering standard precision-recall curves of the pre-trained detector.

Unlike object detection, there are no ground truth bounding boxes for each generated image. We instead use the conditional information (semantic concepts) as ground truth. In an image, a correctly generated object class which aligns with the ground truth is a true positive, any other detections for the same class are false positives. Recall for a class is calculated as the number of correctly generated objects (true positives) divided by total count of that class in the conditions. We train three types of detectors to test the generated samples: objectness detector (OBJ-N), object detector (OBJ-T), and object-attribute detector (OBJ-A). The objectness detector measures whether the generated images contain object-like samples (regardless of object class), the object detector and object-attribute detector measure whether the model generates correct objects or objects with appropriate attributes respectively. In all the following experiment tables, we report accuracy of detectors on ground truth data by standard mAP (bounding boxes based) to verify that the detectors are highly accurate, and our proposed detector score on generated images by models.

**Implementation Details:** We briefly describe our implementation details here. More details can be found in our project repository[2] where we release the code for model training/evaluating and dataset generation. For encoder and decoder, we use residual connection based convolutional blocks. The hidden dimension size is 160 and batchnorm layer is added to stabilize training. For latent size for appearance distribution, we set the latent size to be 64 dimensions with height and width as 16. For location/scale latent distribution, we use 8 dimensions. The learning rate is set as 0.0001 and the model is optimized by Adamax [36]. For baseline models, we use LSTM with 128 hidden dimensions to encode the concept information. In the data generation process, since the sentence-program generation on CLEVR and MNIST is often rule-based, parsing a sentence back can be trivial. We instead directly use the generated programs, which actually lead to more complicated scenes than normal sentences describe.

**Baselines:** We compare our model with two widely used architectures: Conditional DCGAN [37, 38] and Conditional GatedPixelCNN. We first serialize the semantic concepts by traversing the tree, then use LSTM to encode this information. We follow the original proposed DCGAN architecture [38]. For GatedPixelCNN, we use a 10 layer gated PixelCNN [18] with 64 filters, trained by discretized logit loss [39]. Those two models should be able to use LSTM as a powerful global planning model for the distribution based on the semantic conditions. We also use a simple product of experts based VAE model as a baseline[40]. The product of experts simply combines all concepts and generates a text based distribution without considering scene structures.

### 4.1 ColorMNIST dataset

MNIST contains hand-written digits with large variations and is often used for testing generative models. We create a ColorMNIST dataset which contains images with up to two digits in an image. We use 2 size attributes (small, large), 6 color attributes, and 4 relation attributes (top, bottom, left, right) to create compositional scene images. We use 8000 training and 8000 test images.

The results are summarized in Tab. 5b. From objectness detector to object-attribute detector, the metrics are progressively more challenging. We found that all methods can produce images that contain "digit-like" content. PixelCNN generates the images with highest quality pixel detail. PoE-VAE has relatively lower score due to the loss of global scene structure. Our model manages to preserve the most semantic content. Gated-PoE can lead to a 2-3% performance boost. Note that using PoE in operators, which has fewer parameters, can also lead to good generation results.

### 4.2 Model evaluation on CLEVR-G dataset

CLEVR is a standard dataset containing rich compositionality and complicated scenes for examining reasoning in visual question answering. We create a CLEVR-G dataset which contains 10000 64x64

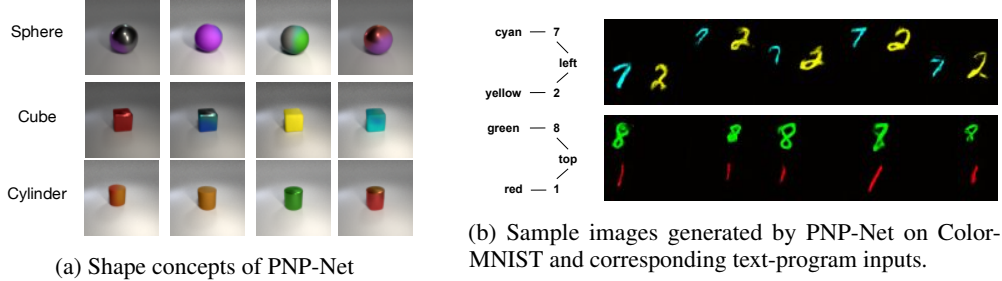

(a) Shape concepts of PNP-Net

(b) Sample images generated by PNP-Net on Color-MNIST and corresponding text-program inputs.

Figure 4: (l): Visualizing the shape concepts in CLEVR-G. (r): Samples in Color-MNIST.

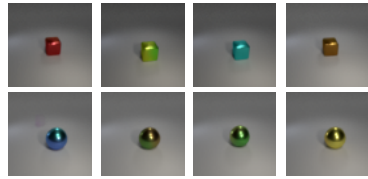

(a) Samples from attribute concept *Metal* grounded on different objects.

| Method | OBJ-N | OBJ-T | OBJ-A |
|---|---|---|---|
| GT | 0.999 | 0.999 | 0.999 |
| DC-GAN | **0.990** | 0.211 | 0.146 |
| PixelCNN | 0.921 | **0.474** | 0.318 |
| PoE-VAE | 0.913 | 0.136 | 0.051 |
| Ours | 0.981 | 0.419 | **0.363** |

(b) Detector scores on Color-MNIST. GT-means we use the exact real image in test setas the input to our pre-trained detector.

Figure 5: Qualitative comparisons between PNP-Net and the baselines on ColorMNIST (left). Visualization of visual concepts defined in our formulation (right).

training images and 10000 testing images. It contains 2 sizes (small, large), 2 materials (rubber, metal), 8 colors, and 6 relations (left, right, left-front, left-behind, right-front, right-behind).

Fig. 6a shows examples of images generated by PNP-Net and baselines. Tab. 6b provides quantitative results. Generally, all methods perform reasonably well at generating a set of objects (high objectness score, qualitatively the correct number of objects). However, our structured prior gives much more accurate depictions of the required objects and their attributes (object type and object-attribute scores, qualitatively correct examples). Further, some global coherence is captured in the generated images (e.g. shape variance with respect to camera distance and position).

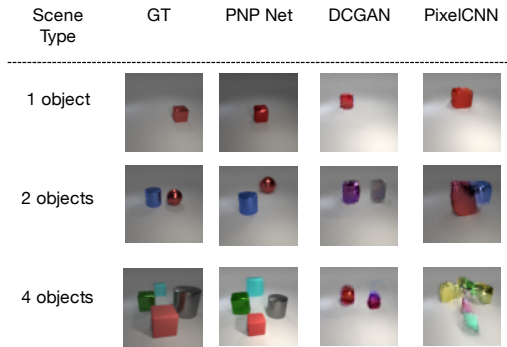

(a) Samples of different models on CLEVR-G dataset.

| Method | OBJ-N | OBJ-T | OBJ-A |
|---|---|---|---|
| GT | 0.999 | 0.998 | 0.976 |
| DC-GAN | **0.979** | 0.566 | 0.176 |
| PixelCNN | 0.894 | 0.444 | 0.074 |
| PoE-VAE | 0.974 | 0.493 | 0.134 |
| Ours | 0.971 | **0.833** | **0.737** |

(b) Detector scores on CLEVR-G dataset. GT means we use the exact real image in test set as the input to our pre-trained detector.

Figure 6: Qualitative (left) and quantitative (right) comparisons on CLEVR-G.

**Visualizing the learned concepts**: Our model learns the primitive concepts in visual words. It would be interesting to see what distributions are learned during training. We visualize the shape concepts that the model is learning by pushing the samples through the decoder. The results are shown in Fig. 4a. Our model successfully learns those concepts separately. We also show the attribute concept generation by grounding it to an object (required by **Describe**). The results are shown in Fig. 5a.

**Zero-shot combination:** As our model is comprised of reusable modules, it potentially has better ability to generalize to unseen combinations. We further create a CLEVR-G-ZS dataset by the following process: We split all concepts into two disjoint sets: colors into $c_1$ and $c_2$, materials into $m_1$ and $m_2$, objects into $o_1$ and $o_2$, and relations into $r_1$ and $r_2$. In our zero-shot setting, the combination of $o_1 \cap m_2 \cap c_2 \cap r_1$ and $o_2 \cap m_1 \cap c_1 \cap r_2$ are omitted in the training set, while the test set only contains them. The model has to exploit its modular nature to handle the generation of unseen combinations. Qualitative results are shown in Fig. 7a and corresponding detector scores are summarize in Table 7b. We use the same pre-trained detector as our non zero-shot experiments. Note that the object-attributes scores under zero-shot setting are not directly comparable with non zero-shot results, since the number of ground-truth object-attribute classes in the zero-shot test set is smaller than that of a normal test set.

**Generalize to complicated scenes:** To test whether our model has the ability to scale up to more complex scenes by assembling reusable modules, we design the following two experiments: (1) We create a 128x128 version dataset with up to 8 objects, then we train and test our model on this more complicated dataset; (2) We create another 128x128 version dataset with up to 4 objects, then we train our model on this up-to-4 dataset and test on the up-to-8 dataset. More complex scenes are intrinsically harder with richer object interactions and occlusions, but the modular property of PNP-Net enables it to properly handle complicated semantics even with easy training data. We summarize the results in Table 1. It shows that our model still outperforms the scores of baseline methods in simpler scenes. Also, our model is robust to the complexity of scenes due to the modularization embedded in our generative models.

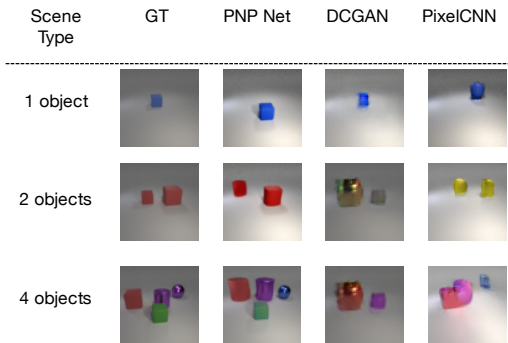

| Method | OBJ-N | OBJ-T | OBJ-A |
|--------|-------|-------|-------|
| DC-GAN | **0.989** | 0.418 | 0.332 |
| PixelCNN | 0.899 | 0.420 | 0.192 |
| PoE-VAE | 0.975 | 0.441 | 0.318 |
| Ours | 0.970 | **0.752** | **0.734** |

(b) Detector scores on CLEVR-G under zero-shot setting. Again, the numbers here are not directly comparable to the ones in Fig. 6b as explained in Sec. 4.2

(a) CLEVR-G Zero-shot

Figure 7: Qualitative (left) and quantitative (right) zero-shot comparisons on CLEVR-G.

| Settings(trained on) | OBJ-N | OBJ-T | OBJ-A |
|----------------------|-------|-------|-------|
| GT(-) | 0.979 | 0.977 | 0.943 |
| Ours (up-to-8) | 0.973 | 0.734 | 0.567 |
| Ours (up-to-4) | 0.976 | 0.715 | 0.518 |

Table 1: Detector scores on CLEVR-G 128x128 images with up to 8 ojbects

# 5 Conclusion

We proposed a novel programmatic approach to constructing priors for generative modeling of complex scenes. A set of modular components can be combined to represent complex abstract concepts. Individual components represent base concepts such as a sphere, or the material property shiny. These are combined via aggregation operators that allow for modification or interactions between components. We demonstrated that these priors can be used to model the variability and compositional aspects of complex images consisting of multiple entities with different properties, outperforming related methods that do not model scene structure.

## Footnotes

[1]In the variational autoencoder setting, PoE potentially has a "ghost information" issue and lacks the ability to control how experts are combined. See the supplementary material for more information.

[2]https://github.com/Lucas2012/ProbabilisticNeuralProgrammedNetwork

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
