[Reviews · NeurIPS 2018]

Reviewer 1



This paper tries to solve the text to scene image generation problem. Technically, it proposes a conditional variant of VAE with a learnable prior distribution. As for scene generation, a concept mapping operator and four aggregation operators is proposed to obtain a distribution of latent variables that carry the semantics given by the tree structured concepts. The operators are compositional and reusable, making it possible to generalize. The methods are compared with popular baseline methods on two real datasets both quantitatively and qualitatively. Generally, the paper is well-written and clearly structured. Pros: 1. The problem of scene generation is of general interests and quite challenging. Though the results are very preliminary, these show the basic ability to composite visual concepts and generate scenes. 2. The experimental settings are detailed and the results support the main claim. Cons: 1. As the PNP-Net uses semantics information and ground truth bounding boxes in training, it’s not proper to call it an unsupervised method. I also suggest that the author state clearly that PNP-Net leverages more ground truth information when comparing with DCGAN and Pixel-CNN. 2. One possible weakness of PNP-Net is the lack of coherence among objects. For instance, in the CLEVR-G dataset, the lighting conditions are shared among different objects, while in the generated scenes, objects often have inconsistent shadow. Nevertheless, I think these are not major issues and the paper make a good contribution towards scene generation. It should be accepted.

Reviewer 2



I have read the other reviews and the authors' rebuttal. My opinion remains that this is a strong submission and should be accepted. I appreciate the authors addressed each of my concerns and questions in sufficient detail. -------------------- This paper proposes a new, modular neural architecture for generating images of scenes composed of attributed objects. The model takes as input a description of the scene to be generated, in the form of a tree-structured 'program' describing the objects, their attributes and their spatial relations. Each elementary concept (object, attribute) is encoded into a distribution over spatial and visual appearance attributes (mu, sigma). These distributions are then composed--following the program tree structure--using a set of operators for combining attributes, describing objects with attributes, and laying out objects in the space. The final result is a final distribution which can be sampled to produce a latent vector representation of the entire scene; this latent representation is then decoded into an image via standard image decoder/generator networks. The model is trained using variational inference. The paper applies this model to generating scenes of colored MNIST digits as well as CLEVR-G scenes, where it often outperforms strong text-to-image baselines under an object-detection-based performance metric. This paper makes a good contribution: taking ideas from Neural Module Networks and related work, used for analyzing and answering questions about images, and applying them to task of generating the images themselves. The proposed architecture is reasonable (though I wish, space permitting, more justifications for design decisions were given). The evaluation is appropriate, and the proposed model clearly produces better samples than alternative approaches. I have some concerns (see below), but overall I am currently in favor of accepting this paper. - - - - - - - I found much of the model to be insufficiently described / under-specified. The level of detail at which it is described is almost sufficient for the main paper. But there are also quite a few details which ought to be provided somewhere (at least in the supplemental) in order for this work to be reproducible: -- Figure 2: What is w_layout? I infer that this is probably an embedding for a spatial relation concept, but this is never defined (or even mentioned anywhere in the paper text) -- Figure 2: I believe (according to Equation 3) that *both* inputs should feed into the "Convs" block, yes? -- Equations 2 and 3: It's not clear to which distributions these operators apply: both z_a and z_s, or just z_a? The way Figure 2 is drawn, it looks like they apply to both. But the f convolution in Equation 3 only makes sense for z_a, so I suspect that this is the case. This could use clearing up, and Figure 2 could be re-drawn to be more precise about it. -- Equation 4: How are bounding boxes / A sampled? Figure 2 shows the sampler as conditioned on z_s, but how, precisely? What's the functional form of the distribution being sampled from? -- Transform: I also don't quite follow how the distribution is resized. The bilinear interpolation makes it sound as if the parameter tensors are resampled onto different WxH resolutions--is this correct? -- Equation 5: The generated bounding boxes don't appear in this equation, nor does w_layout. How is the background canvas placement accomplished? Does this step perhaps use Transform as a subroutine? As it stands, Equation 5 only shows the application of the "Refiner CNN," which is one small part of the overall diagram given in Figure 2. -- Line 172: "we first map the semantic information into programmatic operations": is this process manual or automatic? -- Equation 6: You've written the proposal distribution as q(z); I assume you mean q(z | x)? The form of this encoder is not mentioned anywhere. -- There are other architectural details would would belong in the supplement (e.g. the LSTM encoder in the conditional prior). The detector-based performance measure is a nice idea. I'm wondering, though--do you detect the 'spatial relation' attributes such as 'top' and 'left?' It's not obvious how to do that, if that is something your evaluation takes into account. If not, that should also be mentioned. ColorMNIST: It would be good to see some qualitative results to go along with Table 1. Visualizing the learned concepts: If I understand correctly, your model should also permit you to visualize attribute concepts such as 'shiny?' Do I have that right, and if so, what do these look like? Figure 5: In the zero-shot experiment, which concept combinations were omitted from the training data? Knowing this would help to assess the errors that the different methods are making in the presented images. Generalization to complex scenes: If I understand correctly, this experiment the model's ability to train on and generate larger scenes. Can the model generalize to larger scenes than the ones it was trained on? E.g. if you train a model on scenes with at most 4 objects but test it on scenes with up to 8, how does it behave? I would hope to see some graceful degradation, given the modular nature of the model. Two related works which should probably be cited: (Generating structured scenes from text) "Image Generation from Scene Graphs," Johnson et al. CVPR 2018 (Disentangled representations) "Deep Convolutional Inverse Graphics Network," Kulkarni et al. NIPS 2015

Reviewer 3



I read the authors feedback and they clarified some ambiguities and they mentioned that they will be updating the paper with more details and updating my score based on that. ------ The focus of a paper is text to image generation and they achieve this in the VAE framework. The core part of the model is the compositional generation of latent z by dynamically using reusable aggregator modules. Pros -A very interesting approach in constructing an informative latent code for complicated scences using modular components -Good abstract overview and discussion -Interesting and good results Cons -While the abstract description is great, the model section fails to transmit the ideas clearly. -A lot of details is missing from the model description -No information is given regarding network acthiciteure and training schemes Details: There is a formatting error for the title of Sec 3. Figure 3: What is the difference between dashed and solid lines in the tree? And why transform and describe are merged into single node? Eq 2: I think the gating mechanism is an important aspect that makes the PoE work, but even in appendix it's not quite justified why the gating helps. A figure on a simple toyish example could be beneficial. Each concept consists of two distribution, one for appearance and one for location/scale, are the aggregation operators applied to both elements in parallel? I don't think I quite understand how the tree structure is constructed from a given a description text. Authors mentioned, learnable prior p(z), what is the structure of the network for prior? My understanding is that CLEVER dataset comes with questions, are the authors converting the questions into statements manually? The paper mentions that the proposed method should results in a disentangled latent code, hence it would be very useful to have some experiments showing that is the case. While the included detector scores are very useful, it would be interesting to see more detailed analyses of the learned latent code. The reasoning for the overall score: This paper potentially could be a good paper when some of the details are more clear. Looking forward to hearing authors feedback and then being able to have a better evaluation. The reasoning for the confidence score: As mentioned earlier, due to some ambiguities and missing information I have not quite understood the paper in detail.